# Salt Stress-Induced Modulation of Porphyrin Biosynthesis, Photoprotection, and Antioxidant Properties in Rice Plants (*Oryza sativa*)

**DOI:** 10.3390/antiox12081618

**Published:** 2023-08-15

**Authors:** Anh Trung Nguyen, Lien Hong Tran, Sunyo Jung

**Affiliations:** BK21 FOUR KNU Creative BioResearch Group, School of Life Sciences, Kyungpook National University, Daegu 41566, Republic of Korea; nguyenanh18190@knu.ac.kr (A.T.N.); hong.tran@unimi.it (L.H.T.)

**Keywords:** antioxidant enzymes, chlorophyll and heme biosynthesis, photoprotection, rice, salt stress

## Abstract

Salt stress disrupts cellular ion homeostasis and adversely impacts plant growth and productivity. We examined the regulatory mechanisms of porphyrin biosynthesis, photoprotection, and antioxidant properties in salt-stressed rice seedlings. In response to 150 mM NaCl, the rice seedlings exhibited dehydration, reduced relative water content, and increased levels of conductivity, malondialdehyde, and H_2_O_2_. The expression levels of the salt-stress-responsive genes *NHX1*, *SOS1*, and *MYB* drastically increased after NaCl treatment. The seedlings grown under NaCl stress displayed declines in *F*_v_/*F*_m_, Φ_PSII_, rETR_max_, and photochemical quenching but increases in nonphotochemical quenching (NPQ) and the expression of genes involved in zeaxanthin formation, *BCH*, and *VDE*. Under salt stress conditions, levels of chlorophyll precursors significantly decreased compared to controls, matching the downregulation of *CHLD*, *CHLH*, *CHLI*, and *PORB*. By contrast, NaCl treatment led to increased heme content at 24 h of treatment and significant upregulations of *FC2*, *HO1*, and *HO2* compared to controls. Salt-stressed seedlings also increased their expression of *CATs* (*catalases*) and *APXs* (*ascorbate peroxidases*) as well as the activities of superoxide dismutase, CAT, APX, and peroxidase. Our results indicate that chlorophyll and heme biosynthesis involve the protective strategies for salt stress alleviation through photoprotection by the scavenging of chlorophyll precursors and NPQ as well as activating antioxidant enzymes.

## 1. Introduction

Soil salinity causes an accumulation of sodium (Na^+^) in the shoot and inhibits photosynthesis and growth, resulting in metabolic adaptations and developmental changes [1,2]. Therefore, maintaining a low Na^+^ concentration within cells under salt stress is crucial for plant growth and development [3]. A plasma membrane Na^+^/H^+^ antiporter encoded by *salt overly sensitive 1* (*SOS1*) catalyzes Na^+^ efflux [4,5], while a tonoplast Na^+^/H^+^ antiporter, *At*NHX1, acts by compartmentalizing Na^+^ into the vacuole [6]. Recent studies on the mechanisms of plant salt tolerance have focused on osmoregulation, redox and ionic homeostasis regulation, and signaling-mediated adjustment under salinity stress [1,3,7].

Under salt stress conditions, ion accumulation disrupts the biosynthesis of photosynthetic pigments including chlorophyll in plants [8]. Porphyrin biosynthesis starts with glutamyl-tRNA^Glu^, and 5-aminolevulinic acid (ALA) is metabolized to form tetrapyrroles [9]. In the last common step of the porphyrin biosynthesis pathway, protoporphyrinogen IX is oxidized into protoporphyrin IX (Proto IX) by protoporphyrinogen oxidase (PPO). Fe-chelatase inserts Fe^2+^ into Proto IX to produce heme, while Mg-chelatase (MgCh), which comprises the CHLD, CHLH, and CHLI subunits, inserts Mg^2+^ into Proto IX to produce Mg-Proto IX [9,10]. Heme acts as a cofactor for important antioxidant enzymes involved in detoxifying reactive oxygen species (ROS) [9,11]. In plants, porphyrins are essential in various processes, including light harvesting, energy transfer, detoxification, and signal transduction [9,10]. The regulation of porphyrin biosynthesis under various abiotic stresses, including iron deficiency, chilling, and drought, has been reported in previous studies [12,13,14]; however, its regulatory mechanism under salt stress has not yet been characterized.

Na^+^ inhibits photosynthesis by disrupting chloroplast function [2]. The absorption of more light energy than needed for photosynthesis by the photosynthetic machinery causes the generation of ROS. Photochemical quenching (qP) is used to refer to a decrease in chlorophyll *a* fluorescence resulting from light energy being converted into chemical energy in photosynthesis [15,16]. Nonphotochemical quenching (NPQ) is an important photoprotective mechanism that protects photosynthetic complexes from damage due to excess light. Light-harvesting complexes (LHCs) are known to involve photoprotective quenching to prevent impairment of the photosynthetic systems under excess light [17]. Zeaxanthin, a molecule for photoprotection, also plays a role in the dissipation of excess light energy under stress conditions [17,18,19]. While thermal energy dissipation prevents ROS formation, additional protective mechanisms are available to detoxify ROS.

Various physiological and metabolic processes produce ROS under salt stress, and plants must improve their antioxidant capacity to keep the equilibrium between ROS production and scavenging. Excess ROS are scavenged by antioxidant metabolites such as ascorbate (Asc) and by ROS-detoxifying enzymes [20,21]. The primary scavenger in ROS detoxification in plants is superoxide dismutase (SOD), which converts superoxide (O2^•−^) into H_2_O_2_ and protects cells against oxidative stress [22,23]. Ascorbate peroxidase (APX) catalyzes the elimination of the toxic products of SOD at the expense of oxidizing Asc into monodehydroascorbate. Catalase (CAT) and peroxidase (POD) can also detoxify H_2_O_2_ [17,24].

In this study, we aimed to explore the salt-stress-induced regulation of porphyrin biosynthesis and its role in mitigating the harmful effects of salt stress. First, we examined how plants respond to changes in oxidative stress status by regulating the expression of stress-related genes and adjusting photosynthetic performance and photoprotection in rice seedlings under NaCl stress. We dissected how chlorophyll and heme biosynthesis are regulated to overcome salt stress by analyzing metabolite and gene expression profiles related to porphyrin metabolism. We also determined ROS scavenging capability to assess antioxidant properties under salt stress. Our results indicate that the regulation of chlorophyll and heme biosynthesis is involved in adaptation strategies to alleviate salt-stress-induced oxidative stress by impacting photoprotection and ROS scavenging.

## 2. Materials and Methods

### 2.1. Plant Growth Conditions and Salt Stress Treatments

Rice seeds (*Oryza sativa* cv. Dongjin) were germinated and hydroponically grown in half-strength Hoagland solution under a 14 h light/10 h dark photoperiod with 200 μmol m^−2^ s^−1^ photosynthetic photon flux density (PPFD) for 2 weeks. After 2 weeks of growth, uniformly developed seedlings were selected for salinity stress treatments. For testing plant responses to salt stress, the rice seedlings were transferred into half-strength Hoagland solution without (control) or with 150 mM NaCl (salt stress) under the same light conditions as for the early growth. The young, fully expanded leaves were harvested 24, 48, and 72 h after exposure to NaCl treatment or 24 h after control treatment for further analysis. To measure dry biomass, leaves were dried at 80 °C for 48 h and then weighed. In an independent experiment, each treatment at different time points contained 72 plants.

### 2.2. Relative Water Content

Leaves were excised to measure the relative water content (RWC) and their fresh weight (FW) was measured. The rehydrated weight was measured after immersing them in deionized water at 4 °C overnight. Then, the leaves were dried at 80 °C and their dry weight (DW) was weighed. The RWC was determined as RWC (%) = (FW − DW)/(rehydrated weight − DW) × 100.

### 2.3. Conductivity Measurement

Leaf squares were cut from the salt-treated plants at given time points and floated abaxial side up in miliQ water for 25 min at 25 °C under agitation. Cellular leakage in the bathing solution was determined periodically via the detection of electrolyte leakage using a conductivity meter (Cole-Parmer Instruments, Vernon Hills, IL, USA).

### 2.4. Lipid Peroxidation

For the estimation of lipid peroxidation, the production of malondialdehyde (MDA) was measured using the thiobarbituric acid method with some modifications [25]. Leaf tissues were homogenized in a 0.5% thiobarbituric acid solution in 20% trichloroacetic acid (TCA) and centrifuged. The supernatants were boiled in a water bath for 25 min. The centrifuged supernatants were used for spectrophotometric measurement at 532 nm.

### 2.5. In Vivo Detection of H_2_O_2_

For the visual detection of H_2_O_2_, leaves were cut and incubated in a 1 mg mL^−1^ 3,3-diaminobenzidine (DAB) solution (pH 3.8) for 4 h under light at 25 °C [26]. Then, the leaves were boiled in ethanol for 10 min to terminate the reaction, which decolorized the leaves except for the brown polymerization product. The brown spots indicate the reaction of DAB with H_2_O_2_.

### 2.6. RT-qPCR Analysis

Total RNA was extracted from leaves with TRIzol Reagent (Invitrogen, Carlsbad, CA, USA). The extracted RNA (5 µg) from each sample was used to make cDNA in a reverse transcription reaction (*ImProm-II™* Reverse Transcription System, Promega, WI, USA). Next, qPCR analysis was conducted using cDNA from each sample, gene-specific primers (Appendix A), and Power SYBR™ Green PCR Master Mix (Applied Biosystems, Waltham, MA, USA) in a StepOnePlus™ Real-Time PCR system (Applied Biosystems). The RT-qPCR program included 2 min at 50 °C, 10 min at 95 °C, and 40 cycles of 15 s at 95 °C and 1 min at 60 °C. The *Actin* gene was used as an internal control. The expression level of the control sample as a calibrator was set to 1.

### 2.7. Chlorophyll a Fluorescence Measurement

The chlorophyll fluorescence parameters were measured using a pulse amplitude modulation fluorometer (JUNIOR-PAM, Walz, Effeltrich, Germany). The seedlings from each treatment were adapted in darkness for 30 min. The minimum fluorescence (*F*_o_) was obtained upon excitation with a weak measuring beam from a pulse light-emitting diode, while the maximum fluorescence (*F*_m_) was measured upon a saturating pulse of white light. To evaluate the functional damage to the plants, the *F*_v_/*F*_m_ indicating the PSII activity was used. The Φ_PSII_, which is the quantum yield of electron transport through PSII, was quantified as previously performed by Genty et al. [27]. NPQ was defined as NPQ = (*F*_m_ − *F*_m_′)/*F*_m_′ [28], while qP was defined as qP = (*F*_m_′ − *F*′)/(*F*_m_′ − *F*_o_′) [29]. The rapid light curve (RLC) was measured with a gradual increase in actinic light (0 to 1500 μmol m^−2^ s^−1^). The efficiency of electron transport (α, the initial slope of the light curve), maximum relative ETR (rETR_max_), and light saturation I_k_ (irradiance at the onset of light saturation) were determined according to Platt’s formula [30].

### 2.8. Immunoblot Analysis of LHCB Proteins

For total protein extraction, leaf tissue was extracted with a buffer containing 56 mM Na_2_CO_3_, 56 mM DTT, 12% (*w*/*v*) sucrose, 2% (*w*/*v*) SDS, and 2 mM EDTA, pH 8.0. After centrifugation, the supernatants were taken to obtain the total soluble proteins. The proteins were separated on a 12% SDS-PAGE and electroblotted onto PVDF membranes. The antibodies for LHC proteins of PSII (LHCB) and α-Tubulin were obtained from Agrisera (Agrisera, Vännäs, Sweden). Immunodetection was performed according to standard procedures (Roche, Basel, Switzerland). The density of each band was quantified via ImageJ (National Institutes of Health, Bethesda, MD, USA).

### 2.9. Determination of Porphyrin Contents

The ALA-synthesizing capacity was measured as described [31]. Leaf tissues were incubated in 20 mM phosphate buffer containing 40 mM levulinic acid under irradiation. The homogenized samples were resuspended in 1 mL of 20 mM potassium phosphate buffer (pH 6.9) and centrifuged. Then, 100 μL of ethylacetoacetate was mixed with 500 μL of supernatant. The mixture was boiled for 10 min and cooled. The same volume of modified Ehrlich’s reagent was added, and the absorbance was measured at 553 nm. The chlorophyll content was spectrophotometrically determined as described in Lichtenthaler [32]. For the determination of porphyrin levels, leaves were ground in a acetone:methanol:0.1 N NaOH mixture (10:9:1) and centrifuged at 10,000× *g* and 4 °C [33]. The porphyrin metabolites were separated using high-performance liquid chromatography (HPLC) with a Novapak C_18_ column (4 µm particle size, 4.6 mm × 250 mm, Waters, Milford, MA, USA) and a gradient solvent system from 0.1 M ammonium phosphate (pH 5.8) and methanol (20:80, *v*/*v*) to 100% methanol. The column eluate was detected with a fluorescence detector (2474, Waters). For heme determination, heme was extracted as described previously [34]. The acid-acetone extracts were reextracted with diethyl ether. The protoheme was separated via HPLC with a Novapak C_18_ column (Waters) and a solvent system of ethanol:acetic acid:water (66.5:17:16.5). The column eluate was detected at 402 nm.

### 2.10. Assays for Antioxidant Enzymes

The soluble proteins were extracted from leaf tissues via homogenization buffer (100 mM potassium phosphate buffer (pH 7.5), 1 mM phenylmethylsulfonyl fluoride, 2 mM EDTA, and 1% PVP-40) and centrifuged at 15,000× *g* for 20 min at 4 °C. Equal amounts of protein were electrophoresed on 10% nondenaturing polyacrylamide gels at 4 °C. The isoforms of antioxidant enzymes were stained for the enzyme activity of SOD and APX according to the methods of Rao et al. [35]. The isoforms of CAT and POD were stained for enzyme activity as described previously [36,37].

### 2.11. Ascorbate Content Determination

To determine Asc content, leaf tissues were homogenized in 5% metaphosphoric acid and centrifuged at 13,000× *g* for 15 min. The following extraction and chromatic determination of total Asc and the amount of reduced Asc were performed as described by Law et al. [38]. The absorbance of Asc was determined at 525 nm with a spectrophotometer.

### 2.12. Statistical Analysis

All data are shown as means ± standard error (SE) of nine replicates obtained from three independent experiments. The observed data were analyzed with Duncan’s test (at *p* < 0.05) using SPSS 25 software (SPSS Inc., Chicago, IL, USA).

## 3. Results

### 3.1. Effects of Salt Stress on Oxidative Stress Markers and Transcript Levels of Stress-Related Genes

The 14-day-old rice seedlings were transferred to half-strength Hoagland solution with 150 mM NaCl to induce salt stress. In seedlings treated with NaCl for 48 h, the youngest, fully developed leaves displayed dehydration at their tips compared to control leaves (Figure 1A). The RWC of the NaCl-stressed seedlings was reduced, starting at 92% of the control values and reaching 79% at 72 h (Figure 1B).

Conductivity, which is an indicator of cellular leakage, was measured during the NaCl stress treatment. It continuously increased in NaCl-stressed leaves between 12 and 72 h (Figure 1C), indicating impaired plasma membrane integrity. To reconfirm this observation, we assayed seedlings for MDA content as an assay for lipid peroxidation. MDA level continuously increased in response to NaCl stress, with a 2.6-fold increase at 72 h (Figure 1D). To examine whether salt stress increases ROS level, control and NaCl-treated leaves were incubated with 3,3-DAB for detecting H_2_O_2_ generation. The leaves from NaCl-stressed seedlings exhibited dramatically increased H_2_O_2_ generation at 48 and 60 h after salt stress, whereas control leaves did not display any noticeable H_2_O_2_ generation (Figure 1E).

The effect of salt stress on the expression levels of genes involved in salt stress tolerance and photosynthesis was examined. The expression levels of *NHX1*, a vacuolar Na^+^/H^+^ antiporter that sequestrates Na^+^ into vacuoles [6], continuously increased in response to 150 mM NaCl, with an 8.2-fold increase at 72 h (Figure 2A). The expression of *SOS1*, which encodes a plasma membrane Na^+^/H^+^ antiporter catalyzing Na^+^ efflux [4], was drastically upregulated at 24 h after NaCl stress and downregulated at 48 and 72 h, although it remained higher than in untreated controls. The expression of *MYB2*, which encodes a stress-responsive MYB transcription factor [39], was also drastically upregulated at 24 h after salt stress but downregulated at 72 h. The expression levels of *RBCS* encoding the small subunit of Rubisco as well as *LHCB*, the gene encoding LHCs of PSII, were analyzed as markers of photosynthetic gene expression patterns in NaCl-treated seedlings. The transcript levels of *LHCB1*, *LHCB6*, and *RBCS* gradually decreased in response to NaCl stress compared to untreated controls (Figure 2B). The transcript levels of *BCH*, encoding β-carotene hydroxylase (BCH), which converts β-carotene to zeaxanthin in carotenoid biosynthesis [40], were increased by about 64% after 72 h of NaCl treatment compared to the control (Figure 2C). The seedlings grown under salt stress also displayed an upregulated expression of *VDE*, encoding violaxanthin de-epoxidase (VDE), which forms zeaxanthin from violaxanthin [19], compared to the control (Figure 2C).

### 3.2. Effects of Salt Stress on Photosystems, Photosynthetic Efficiency, and Photoprotective Mechanisms

We examined how rice seedlings treated with salt stress develop photoprotective mechanisms to optimize their photosynthesis. In the assessment of leaf photosynthetic capacity using the RLC approach, rETR of PSII increased with the intensity of irradiation (Figure 3A). The rETR under NaCl treatment reached a steady state faster, particularly after 72 h, while the rETR in controls reached a steady state at higher light levels. rETR_max_ and I_k_ in the NaCl-treated seedlings were significantly lower than under control treatments (Table 1). The value of α, the initial slope of the light response, was lower in NaCl-stressed plants than in control plants, with the lowest value in NaCl-treated plants at 72 h. Next, we quantitatively analyzed the LHCB proteins of PSII to evaluate the altered organization of the photosynthetic apparatus under NaCl stress. The levels of the major components of light-harvesting antennae, LHCB1 and LHCB6, were similar between control and NaCl-treated samples at 24 h but began to decrease from 48 h onward (Figure 3B).

The impact of salt stress on photosynthetic performance was also corroborated by measuring changes in *F*_v_/*F*_m_, which is the photochemical quantum efficiency of PSII. The values of *F*_v_/*F*_m_ remained constant until 24 h after NaCl treatment and decreased by 13% and 38% at 48 h and 72 h, respectively, compared to the control (Figure 3C). The seedlings grown under NaCl treatment showed a gradual decline in Φ_PSII_ under 72 h of salt stress compared to control seedlings (Figure 3D). In the rapid estimation of quenching mechanisms under salt stress, NaCl-treated seedlings showed lower qP values compared to control seedlings, with their lowest at 72 h (Figure 3E). Nonradiative energy dissipation through NPQ was higher by 30% and 49% in seedlings treated with NaCl for 24 h and 48 h, respectively, compared to the control treatment (Figure 3F). By contrast, the NPQ level was lower by 40% in NaCl-treated seedlings at 72 h compared to control seedlings.

### 3.3. Salt Stress-Mediated Changes in the Regulation of Chlorophyll and Heme Biosynthesis

The levels of chlorophylls, the end products of the Mg-porphyrin branch in porphyrin biosynthesis, in NaCl-treated seedlings exhibited a 19% decrease at 48 h and 34% at 72 h compared to control seedlings (Figure 4A). A lower synthesizing capacity for ALA was observed in NaCl-stressed seedlings than in control seedlings, with a 78% decrease at 72 h (Figure 4B). The Proto IX levels gradually decreased under 72 h of NaCl stress treatment (Figure 4C). Next, we examined the influence of salt stress on metabolic flux within the Mg branch. The Mg-Proto IX, Mg-Proto IX ME, and Pchlide levels were comparable to control seedlings at 24 h of NaCl treatment, while they began to decrease from 48 h on (Figure 4C).

To explore the molecular mechanisms underlying salt-stress-induced alterations in porphyrin metabolism, we analyzed expression levels for key biosynthetic genes. In the common pathway, the expression levels of *HEMA1* and *GLUTAMATE 1-SEMIALDEHYDE AMINOTRANSFERASE* (*GSA*), encoding enzymes for ALA-synthesizing activity, continuously decreased upon NaCl treatment compared to the control treatment (Figure 5A). The expression levels of *ALAD*, encoding ALA dehydratase, and *PPO1* began to decrease from 48 h onward after NaCl treatment compared to the control treatment. To explain the changes in Mg-porphyrin levels upon salt stress, we next assessed the expression of genes belonging to the chlorophyll branch. *CHLD* began to decrease at 48 h after NaCl treatment, while *CHLH* and *CHLI*, encoding the H and I subunits of MgCh, respectively, were downregulated in NaCl-stressed seedlings, the most strongly at 72 h (Figure 5B). The NaCl treatment downregulated *PROTOCHLOROPHYLLIDE OXIDOREDUCTASE B* (*PORB*) expression, reaching about 16% of the control value at 72 h.

In the heme branch, *Fe-chelatase 2* (*FC2*) was upregulated by 2.2-fold at 24 h after NaCl treatment and returned to control levels at 48 h (Figure 6A). *HO1*, encoding heme oxygenase 1 (HO1), which converts heme to biliverdin IXα [41], was upregulated by up to 4.1-fold in NaCl-treated seedlings at 48 h and returned to control levels at 72 h (Figure 6A). The related gene *HO2* transcript levels also displayed about a twofold increase in the NaCl-treated seedlings. The heme content, the product of the Fe-porphyrin branch, increased by 39% at 24 h and later returned to control levels (Figure 6B).

### 3.4. Activity and Gene Expression Level of ROS-Scavenging Enzymes under Salt Stress

Plants have evolved different ROS-scavenging systems to reduce cellular oxidative stress. NaCl treatment influenced the isozyme profiles and activities of ROS-detoxifying enzymes. The total SOD activities were higher in NaCl-treated seedlings compared to control seedlings, with the greatest increases for the SOD isozymes 1, 2, and 3 at 72 h (Figure 7A). The staining activities of CAT isozymes 2 and 3 strongly increased in response to NaCl treatment compared to untreated controls (Figure 7B). Marked increases in APX staining activities were observed in seedlings grown under NaCl stress, with the greatest increase at 48 h (Figure 7C). NaCl-treated seedlings exhibited higher activities of POD isozymes 2, 3, 4, and 5 compared to controls (Figure 7D).

The expression levels of the genes encoding H_2_O_2_-detoxifying enzymes, APX and CAT, corresponded to their enzyme activities in NaCl-treated seedlings. The expression of *CATa* exhibited a twofold upregulation in seedlings treated with NaCl for 48 h compared to the control treatment (Figure 8A). The NaCl-treated seedlings exhibited a gradual increase in the expression level of *CATb* compared to the control, with a 12-fold increase at 72 h. The highest expression of *CATc* was observed in seedlings treated with NaCl for 24 h and 48 h. In comparison to controls, the expression of *APXa* continuously increased by 1.6, 3, and 4-fold at 24, 48, and 72 h after NaCl treatment, respectively. The transcript levels of *APXb* increased by 3.7-fold at 24 h and decreased at 48 and 72 h, remaining higher than control levels (Figure 8B). To test whether NaCl stress modified the cellular redox metabolism, we assessed Asc content as a marker of oxidative stress. In response to salt stress, the redox state of Asc, which is estimated as Asc × 100/dehydroascorbate (DHA) + Asc, increased slightly at 24 h and decreased at 72 h compared to the control (Figure 8C).

## 4. Discussion

We demonstrate the possible involvement of salt-stress-induced modulation of chlorophyll and heme biosynthesis in plants. Salt stress induced by the presence of NaCl in soil solution limits the absorption of water by plants [8]. In response to 150 mM NaCl stress, rice seedlings exhibited dehydrated leaf tips, a typically visible symptom of salt stress (Figure 1A), and a reduced RWC of 8% and 21% after 48 h and 72 h of prolonged salt stress, respectively, indicating that the plants suffered from mild and severe salt stress (Figure 1B). Marked increases in conductivity and MDA in NaCl-treated seedlings (Figure 1C,D) indicate impaired membrane integrity. Lipid-peroxidation-induced membrane disruption destroys cellular compartments, causes dehydration and loss of solutes, and leads to cell death [42]. The accumulation of oxidatively damaged lipids and greatly increased H_2_O_2_ production 48 and 60 h after NaCl treatment (Figure 1E) indicate increased oxidative stress in salt-stressed plant tissues. The enhanced production of H_2_O_2_ and MDA has been used as the salt stress indicator in various plant species [43,44,45].

Plants have developed the SOS pathway, salt-responsive gene expressions, and transcriptional cascades as protective strategies to alleviate salt stress [5,45,46]. NaCl stress greatly upregulated the expression of *NHX1* and *SOS1* (Figure 2A), key genes regulating Na^+^ sequestration/efflux and protecting cells from Na^+^ toxicity in Arabidopsis and rice [5,44]. MYB, which has a regulatory role in plant tolerance to salt, cold, and dehydration stress [39], increased significantly in NaCl-stressed seedlings at 24 h compared to the control treatment (Figure 2A). The increased expression of *OsMYB2* and *AtMYB* upon salt stress was also observed in other studies [39,47]. These results show a regulation of Na^+^ homeostasis as the primary cellular response to avoid accumulating toxic salt in NaCl-treated plants.

Chloroplasts can initiate signaling that downregulates the expression of nuclear genes through retrograde signaling under stress conditions [48,49]. Although the expression levels of the nuclear-encoded photosynthetic genes PSII, *LHCB1*, and *LHCB6* began to decrease at 24 h after NaCl exposure, the levels of LHCB1 and LHCB6 proteins, major components of light-harvesting antennae, showed a significant decrease at 72 h (Figure 2B and Figure 3B), indicating an impaired structure of the photosynthetic apparatus under salt stress. Next, we examined whether the altered PSII structure results in changes in PSII photochemistry via RLC and the fluorescence characteristics under salt stress. Compared to controls, the lower values of rETR_max_, I_k_, and α in NaCl-treated seedlings (Figure 3A) indicate the decreased utilization of the absorbed light energy for photochemistry. The PSII activity, as indicated by Φ_PSII_ and *F*_v_/*F*_m_, greatly decreased in NaCl-treated seedlings at 48 and 72 h compared to controls (Figure 3C,D), indicating an impairment of photosynthesis as a result of NaCl stress. The decrease in PSII function may result from photodamage due to the decreased excitation dissipation and overreduction of the electron transport system under salt stress. Other studies also demonstrated a general decrease in photochemical parameters (*F*_v_/*F*_m_, Φ_PSII_, and q_P_) under salt stress conditions [50,51].

Photochemical quenching, qP, indicates the onset of photoinhibition in PSII [52]. Lower qP values in NaCl-treated seedlings than in control seedlings (Figure 3E) imply a significant increase in the proportion of closed PSII reaction centers [27]. The decreased qP also suggests an increase in the excitation pressure on PSII [53], which would cause further damage to PSII. In response to potential photodamage, photoprotective mechanisms are vital for the existence of photosynthesis [17,54]. During the early stages of salt stress, non-radiative energy dissipation through NPQ increased by 30% and 49% at 24 h and 48 h, respectively (Figure 3F), indicating increased photoprotection. However, the decline in NPQ by 40% at 72 h suggests a defective thermal dissipation in the late stages of salt stress. The aggregation of LHCII is proposed to be involved in quenching because the LHCII of PSII is directly involved in NPQ [55]. At 72 h after NaCl treatment, the degradation of PSII, as indicated by the lower LHCII level, appears to be related to a decreased NPQ level (Figure 3B,F). Similarly, *Eugenia myrtifolia* plants showed an increase and a decrease in NPQ after mild and severe salt treatment, respectively [56]. Spinach plants reduced photochemical capacity slightly and kept the cell osmotic balance through the production of compatible solutes under mild salt stress, whereas they were unable to respond with photochemical and osmotic adaptation under severe salt stress [57]. Our results indicate that photochemical capacity and NPQ may succumb to severe salt stress. Zeaxanthin can dissipate excess excitation energy as heat in the PSII antenna by a process termed NPQ [19]. The expression levels of *BCH* and *VDE*, genes involved in zeaxanthin biosynthesis [19,40], were noticeably increased in response to NaCl treatment (Figure 2C), thus contributing to photoprotection by sustaining the dissipation of excess energy at PSII.

Next, we examined the metabolic control of chlorophyll biosynthesis under salt stress conditions. In NaCl-treated seedlings, the downregulation of *GSA* and *HEMA1*, genes involved in ALA synthesis [9], matched the gradual decrease in ALA-synthesizing capacity by up to 78% compared to control seedlings (Figure 4B and Figure 5A), showing the salt-stress-induced regulation of ALA biosynthesis at the transcriptional level. The levels of ALA also decreased in etiolated *Brassica campestris* seedlings treated with salt stress [58]. By contrast, salt stress increased ALA levels in cucumber seedlings [59]. During salt stress, the downregulation of *ALAD* and *PPO1*, genes encoding enzymes converting ALA to Proto IX, appears to result in decline in Proto IX, a common precursor for the chlorophyll and heme branches (Figure 4C and Figure 5A). The chlorophyll branch begins with the synthesis of Mg-Proto IX by MgCh [9,10]. Although chlorophyll levels decreased by only 34% at 72 h after NaCl stress, the levels of Mg-Proto IX, Mg-Proto IX ME, and Pchlide, which are chlorophyll precursors, greatly decreased, with a maximal decrease by 84–93% at 72 h compared to controls (Figure 4). The observed changes in metabolic flux within the *Mg branch* under salt stress match the transcript levels of *CHLD*, *CHLH*, *CHLI*, and *PORB* (Figure 5B), indicating that the decrease in Mg-porphyrins results partially from the downregulation of these genes. Other abiotic stresses, including drought, chilling, and iron deficiency stress, also affect the chlorophyll biosynthetic pathway by reducing metabolite levels [12,13,14]. The scavenging of phototoxic chlorophyll precursors in NaCl-treated seedlings seems to be essential to cope with their excited-state dynamics, thereby mitigating salt-stress-induced oxidative damage. By contrast, the downregulation of chlorophyll biosynthesis may also lead to an adverse impact on the photosynthetic apparatus, as indicated by the decreased levels of chlorophyll and LHCB proteins, which are essential components of photosystems, under salt stress conditions.

Heme is an important porphyrin in plants since it is a cofactor for many enzymes [60,61,62]. In the Fe-porphyrin branch, the increase in heme contents by 39% after 24 h of NaCl treatment is likely due to the upregulation of *FC2* by 2.2-fold (Figure 6), suggesting that *FC2* is involved in the salt stress response. The increased level of heme in salt-stressed cucumber [59] is in support of our findings. By contrast, salt stress decreased the levels of heme in etiolated *B. campestris* seedlings [58]. Arabidopsis overexpressing *AtFC1* displayed resistance to high salinity [63], and the overexpression of *Bradyrhizobium japonicum FC* in transgenic rice led to increased heme and improved protection to photodynamically induced oxidative stress [64]. However, both heme content and *FC2* expression levels returned to control levels in the late stages of salt stress (Figure 6). HO is the rate-limiting enzyme in plants and animals, catalyzing the oxidative conversion of heme to biliverdin IXα which is a potent antioxidant [41]. *HO1* and *HO2*, the two genes encoding HO, were significantly upregulated by up to 4.1 and 2-fold, respectively, in NaCl-treated seedlings compared to controls (Figure 6A). The high demand for the antioxidant molecule biliverdin IXα under salt stress may increase *HO* expression. HY1 (HO1 sub-family in Arabidopsis) is suggested to play an important role in salt acclimation signaling [65]. Expression of a *B. napus* HO enhances plant tolerance to mercury-induced stress [61], showing an important role of HO in various environmental stresses. During prolonged salt stress, the sustained levels of *FC2* and heme, which may ensure the cofactor supply for heme proteins, and the upregulation of *HOs* indicate strong involvement of the Fe-porphyrin branch in the salt stress response.

To detoxify ROS, scavenging depends on highly efficient enzymatic components in plant cells [21,66]. ROS overproduced under salt stress conditions are donated to O_2_, thus creating O_2_^•−^ that can be converted to H_2_O_2_ by SOD [22,23]. NaCl stress greatly increased the activities of SOD and POD compared to controls, with a maximal increase at 72 h (Figure 7A,D). H_2_O_2_-scavenging enzymes with heme as a cofactor, including APX, CAT, and POD, are also involved in the plant response to abiotic stress [61,64]. In NaCl-treated seedlings, the great increase in the activities of CAT isozymes 2 and 3 corresponds to the increased transcript levels of *CATa*, *CATb*, and *CATc* (Figure 7 and Figure 8). In parallel with increased APX activities, NaCl-treated seedlings responded to salt stress by upregulating transcript levels of *APXa* and *APXb* (Figure 7 and Figure 8). The increased activity of POD, CAT, and APX in salt-stressed plants can be seen as strengthening the ability of the leaves to decompose H_2_O_2_, thereby mitigating the oxidative damage caused by salt stress. The increased activities of antioxidant enzymes, such as SOD, APX, and CAT, have also been observed with increasing NaCl concentrations in *Chenopodium quinoa* seedlings [67] and in salt-tolerant cultivars of tomato, pea, *Jatropha curcas*, and *Calendula officinalis* [68,69,70,71].

A shift in the Asc/DHA ratio is crucial to transmit abiotic stress signals in plants [72,73]. In our study, the redox state of Asc, as a marker for cellular redox metabolism, slightly increased at 24 h and decreased at 72 h in NaCl-treated seedlings compared to controls (Figure 8C). The alteration in the redox state of Asc in salt-stressed plants accounts for the stress perception of plant cells. Upon salinity stress, the Asc redox status did not significantly change in the leaves of maize seedlings but decreased in the roots [43]. Salt-stress-induced adjustments in the Asc redox state seem to be affected by many factors. Ascorbate is the cofactor of VDE and helps in forming zeaxanthin, thereby participating in NPQ protecting PSII from photoinhibition [74,75]. In the salt-stressed seedlings, the highest upregulation of VDE at 24 h corresponded to the transient increase in the redox state of Asc (Figure 2C and Figure 8C), indicating that Asc may be important in chloroplast protection under salt stress conditions.

## 5. Conclusions

Plants treated with NaCl showed increased oxidative stress and impaired membrane integrity, followed by increased expression of genes involved in Na^+^ efflux and sequestration. Under salt stress, the declines in chlorophyll and LHC proteins, which are essential components for maintaining the integrity of the photosynthetic apparatus, indicate reductions in light harvesting and photosynthetic capacity. The scavenging of phototoxic chlorophyll precursors, partly due to the downregulation of biosynthetic genes, in NaCl-treated seedlings appears important in mitigating the oxidative damage of PSII. NaCl-stressed plants increased NPQ and upregulated the genes related to the dissipation of excess energy, indicating increased photoprotection. The increases in heme and expression levels of *FC2* and *HOs* in NaCl-treated seedlings may contribute to a major role in alleviating salt stress through ensuring the supply of the cofactor for heme proteins. NaCl-stressed plants also developed an enzymatic ROS-scavenging system through greatly stimulated activities of POX, CAT, and APX, which may relate to increased heme levels, to detoxify H_2_O_2_ during salt stress.

Our results demonstrate that differential regulation of chlorophyll and heme biosynthesis can contribute to protecting plants from salt-induced oxidative damage. The protective strategies for salt stress alleviation involve the modulation of the porphyrin biosynthetic pathway through not only photoprotection by scavenging chlorophyll precursors and influencing NPQ, but also activating antioxidant properties by providing the heme cofactor and upregulating HOs. Our study provides new insight into the possible role of the porphyrin biosynthetic pathway in mitigating the harmful effects of salt stress in plants.

## Figures and Tables

**Figure 1 antioxidants-12-01618-f001:**
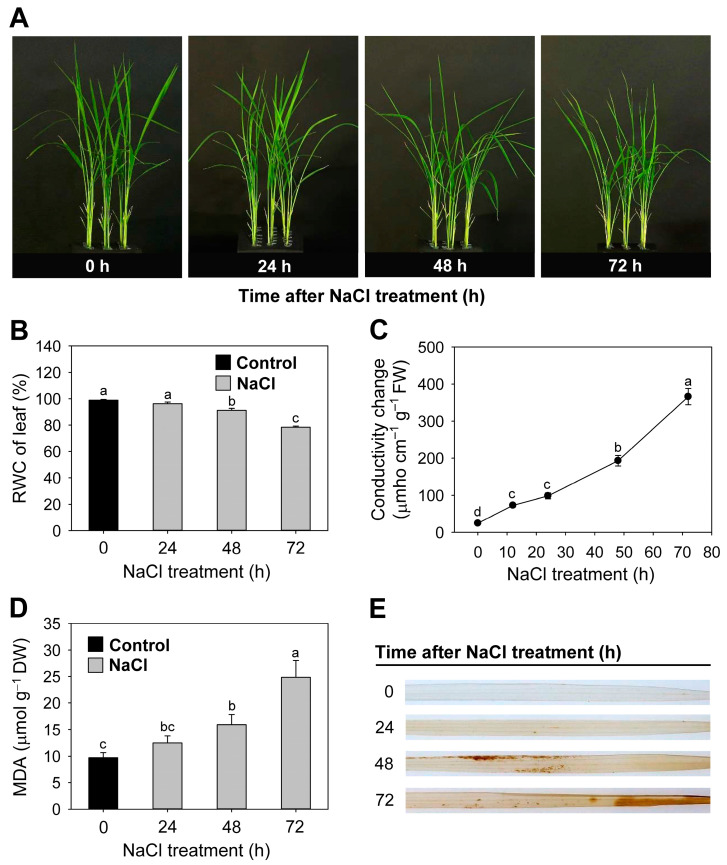
Oxidative metabolism in leaves of rice seedlings exposed to salt stress. (**A**) Phenotypes associated with NaCl-stress-related symptoms. (**B**) RWC of leaves. (**C**) Conductivity change. (**D**) MDA levels. (**E**) H_2_O_2_-DAB-staining in leaves. Brown spots indicate H_2_O_2_ localization. The 14-day-old rice seedlings were transferred to half-strength Hoagland solution without or with 150 mM NaCl for 3 days. Control; half-strength Hoagland solution. 0, 24, 48, and 72; 0, 24, 48, and 72 h after NaCl treatment, respectively. Data are means ± SE of nine replicates obtained from three independent experiments. Mean values followed by different lowercase letters are significantly different at *p* < 0.05 by Duncan’s test. DW: dry weight.

**Figure 2 antioxidants-12-01618-f002:**
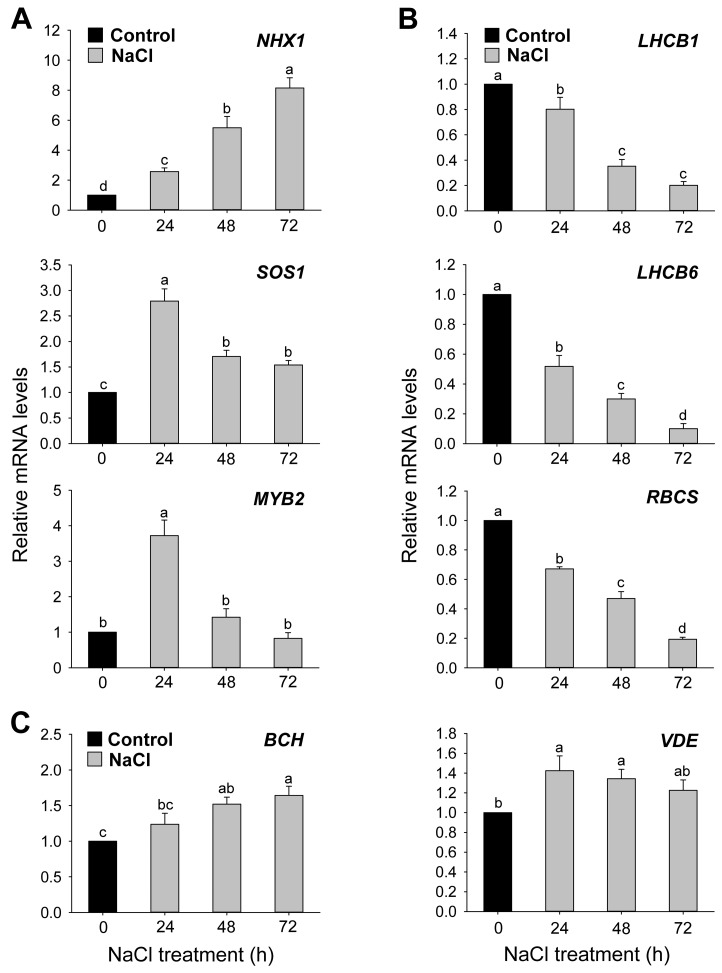
Expression of stress-responsive genes induced by salt stress in leaves of rice seedlings. (**A**) Salt-stress-responsive genes. (**B**) Photosynthetic genes. (**C**) Genes involved in zeaxanthin formation. *Actin* was used as an internal control. The control sample was used as a calibrator, with its expression level set to 1. Data are means ± SE of nine replicates obtained from three independent experiments. Mean values followed by different lowercase letters are significantly different at *p* < 0.05 by Duncan’s test.

**Figure 3 antioxidants-12-01618-f003:**
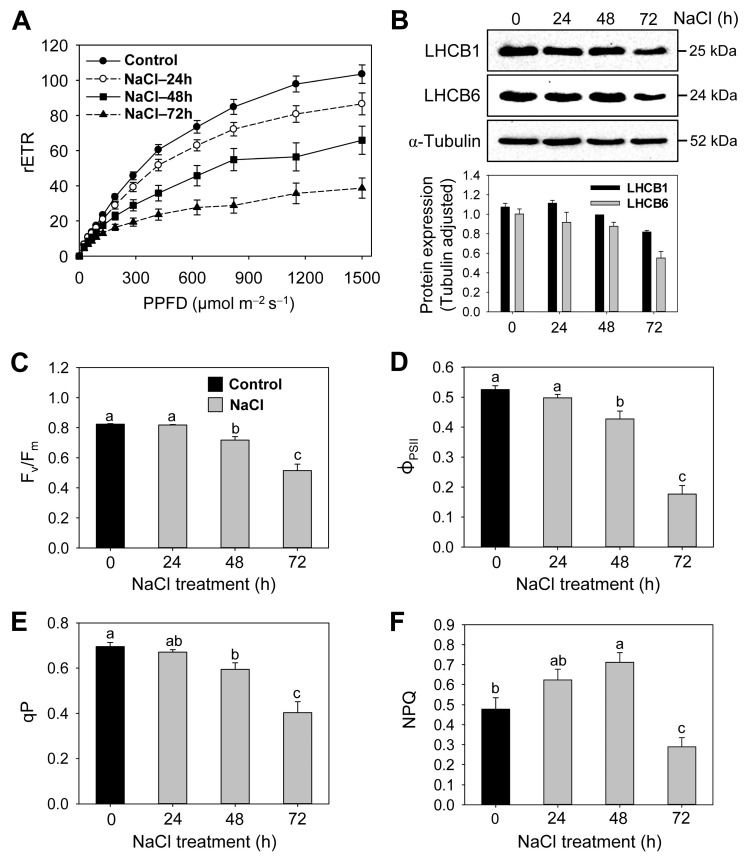
Effects of salt stress on structure and function of photosystems and chlorophyll fluorescence quenching parameters in rice seedlings. (**A**) RLCs of the rETR. (**B**) Immunoblot analysis of the light-harvesting chlorophyll-binding proteins of PSII. Alpha-tubulin was used as a loading control. The relative expression levels of the LHCB proteins were quantified using ImageJ and normalized to Tubulin. (**C**) *F*_v_/*F*_m_. (**D**) Φ_PSII_. (**E**,**F**) Quenching parameters, qP and NPQ. Data are means ± SE of nine replicates obtained from three independent experiments (except six replicates from three independent experiments for immunoblot analysis). Mean values followed by different lowercase letters are significantly different at *p* < 0.05 by Duncan’s test.

**Figure 4 antioxidants-12-01618-f004:**
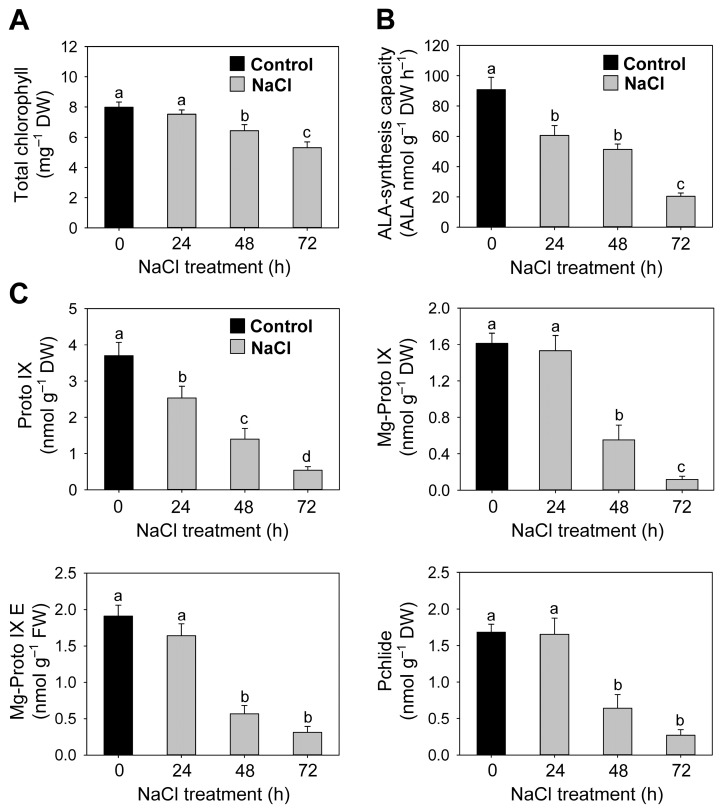
Effects of salt stress on metabolic intermediates of the common and chlorophyll branches in the porphyrin biosynthetic pathway in leaves of rice seedlings. (**A**) Chlorophyll. (**B**) ALA-synthesizing capacity. (**C**) Proto IX and Mg-porphyrin intermediates. Data are means ± SE of nine replicates obtained from three independent experiments. Mean values followed by different lowercase letters are significantly different at *p* < 0.05 by Duncan’s test.

**Figure 5 antioxidants-12-01618-f005:**
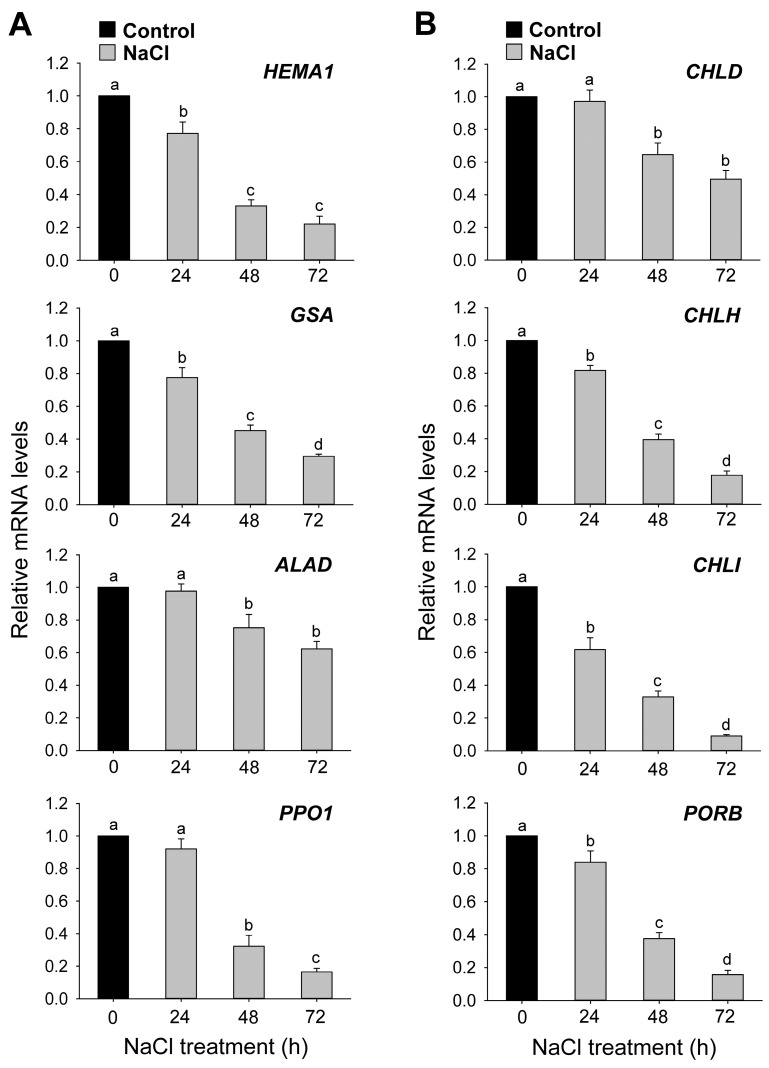
Salt-stress-induced changes in the expression of genes encoding enzymes of the chlorophyll biosynthetic pathway in leaves of rice seedlings. (**A**) Common branch. (**B**) Chlorophyll branch. *Actin* was used as an internal control. The control sample was used as a calibrator, with its expression level set to 1. Data are means ± SE of nine replicates obtained from three independent experiments. Mean values followed by different lowercase letters are significantly different at *p* < 0.05 by Duncan’s test.

**Figure 6 antioxidants-12-01618-f006:**
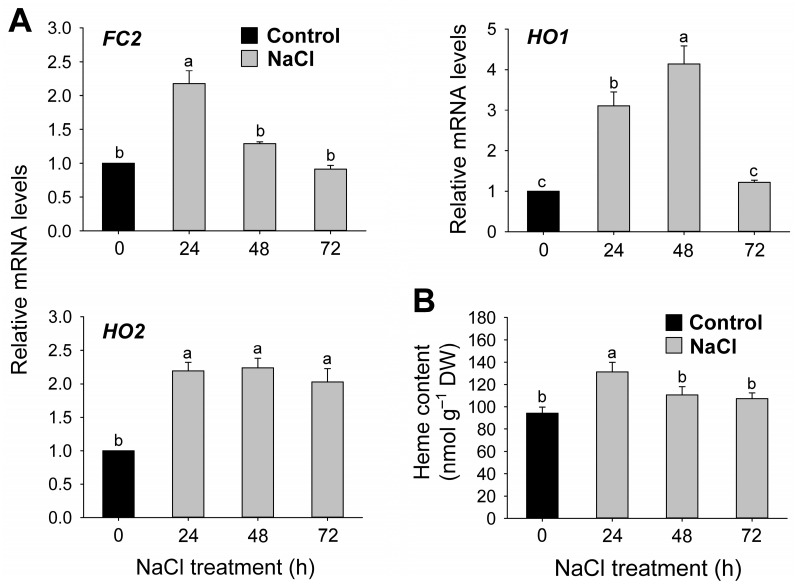
Effects of salt stress on the expression of genes encoding enzymes of the heme branch and heme content in leaves of rice seedlings. (**A**) Expression of genes in the heme branch. (**B**) Heme content. *Actin* was used as an internal control. The control sample was used as a calibrator, with its expression level set to 1. Data are means ± SE of nine replicates obtained from three independent experiments. Mean values followed by different lowercase letters are significantly different at *p* < 0.05 by Duncan’s test.

**Figure 7 antioxidants-12-01618-f007:**
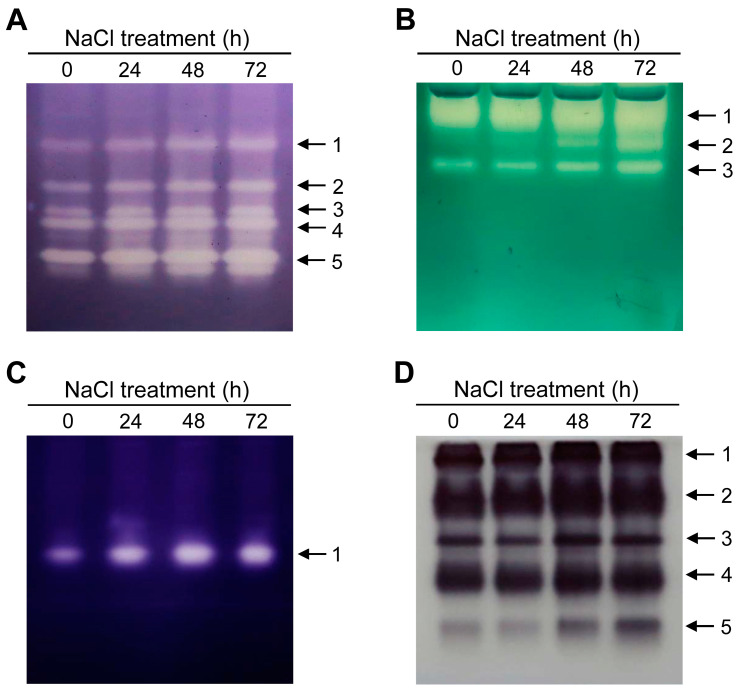
Isozyme profiles of antioxidant enzymes in rice seedlings grown under salt stress (**A**) SOD. (**B**) CAT. (**C**) APX. (**D**) POD. The plants were subjected to the same treatments as in Figure 1. The numbers show each isozyme of antioxidant enzymes in order of bands from the top.

**Figure 8 antioxidants-12-01618-f008:**
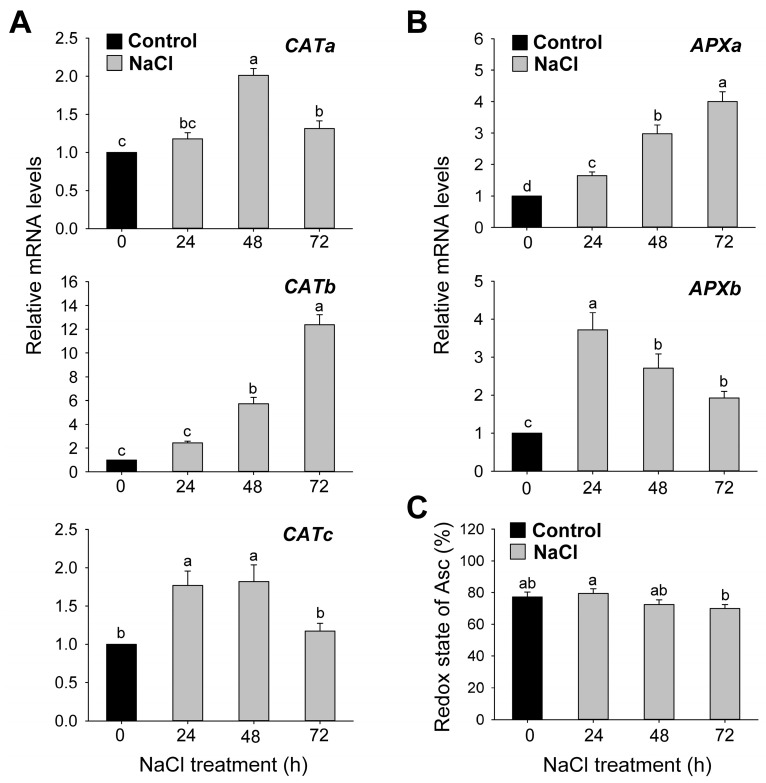
Effects of salt stress on the expression of genes encoding the H_2_O_2_-scavenging enzymes and redox state of ascorbate. (**A**) *CATs*. (**B**) *APXs*. (**C**) Redox state of ascorbate: estimated as Asc × 100/Asc_t_, where Asc_t_ = DHA + Asc. *Actin* was used as an internal control. The control sample was used as a calibrator, with its expression level set to 1. Data are means ± SE of nine replicates obtained from three independent experiments. Mean values followed by different lowercase letters are significantly different at *p* < 0.05 by Duncan’s test.

**Table 1 antioxidants-12-01618-t001:** Effects of salt stress on RLC parameters, rETR_max_, α, and I_k_ in leaves of rice seedlings.

Treatment	rETR_max_	α	I_k_ (μmol m^−2^ s^−1^)
Control	103.5 ± 5.7 ^a^	0.169 ± 0.010 ^a^	634 ± 53 ^a^
NaCl—24 h	85.3 ± 5.4 ^b^	0.157 ± 0.016 ^a^	600 ± 67 ^a^
NaCl—48 h	71.1 ± 8.6 ^b^	0.127 ± 0.012 ^b^	562 ± 52 ^b^
NaCl—72 h	36.5 ± 5.6 ^c^	0.091 ± 0.011 ^c^	467 ± 88 ^b^

Data are means ± SE of nine replicates obtained from three independent experiments. Within each column, mean values followed by different lowercase letters are significantly different at *p* < 0.05 by Duncan’s test.

## Data Availability

The data are contained within the article.

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
