# Peer review of "Salt Stress-Induced Modulation of Porphyrin Biosynthesis, Photoprotection, and Antioxidant Properties in Rice Plants (Oryza sativa)"

_antioxidants, 2023, doi:10.3390/antiox12081618_

Round 1

Reviewer 1 Report

In this manuscript, the authors conducted an interesting study on how salt stress may be involved in the modulation of heme and chlorophyll biosynthesis as defensive response of plants to salt stress itself.

As the authors well indicate, the most direct effects generated by salt stress on the plant are: Na+ toxicity; Osmotic stress that causes plant difficulty in absorbing water resulting in foliar dehydration and a general decrease in photochemical parameters (Fv/Fm, ΦPSII, and qP) with impaired photosynthesis.

The authors focused their study, more on the latter aspect.

The experimentation was well performed and the methods adopted are reliable and effective.

The results provided by the authors are particularly useful for knowledge, on redox regulation and ion homeostasis.

Faced with the breadth and comprehensiveness of the results in the areas of photochemistry, photoprotection, and activation of antioxidant properties in mesophyll tissue, I can only add a few observations.

Why did the authors not perform a direct analysis of ROS in mesophilic tissue?

Many commercial tests and kits are available.

Perhaps because the reagents used in commercial kits are highly context-dependent and not all ROS interact similarly?

However, a marked increase in MDA indicates lipid peroxidation induced by strongly oxidizing chemical species as evidenced by an increase in H2O2 at 48 and 60 hours after NaCl treatment.

I invite the authors to clarify a general concept; Since homeostatic modulation in defense of salt stress, at the photochemical level, is essentially manifested by an increase in NPQ, in the first 48 h while it begins to decrease at 78 h, is it possible to speak of an adaptive response only under mild stress conditions and not under severe stress?

I invite the authors to establish a homeostatic parallelism of mild salt stress in photochemical processes with significant reduction of ROS and in osmoregulatory processes, where the accumulation of NaCl in vacuoles, decreases the water potential by allowing greater water recall and itself acting as an osmolyte.

  • See Di Martino et al.2003 New Phytologist; section Osmolytes distribution in spinach leaf cells

Under severe salt stress conditions, the plant is unable to respond with either photochemical or osmolytic adaptive systems.

Although the authors do not provide data on stomatal conductivity, probably under conditions of severe stress and reduced stomatal rhyme, the initiation of pseudocyclic processes with ROS formation in a photorespiratory metabolic context places the plant in a state of total physiological succumbing in response to salt stress.

Acceptable and understandable

Author Response

Response to Reviewer 1 Comments

Point 1: Why did the authors not perform a direct analysis of ROS in mesophilic tissue? Many commercial tests and kits are available. Perhaps because the reagents used in commercial kits are highly context-dependent and not all ROS interact similarly? However, a marked increase in MDA indicates lipid peroxidation induced by strongly oxidizing chemical species as evidenced by an increase in H2O2 at 48 and 60 hours after NaCl treatment.

Response 1: We measured changes in H2O2 production by DAB-staining to evaluate oxidative stress in NaCl-treated seedlings, since the increased production of H2O2 and MDA is commonly used as the salt-stress indicator. As the reviewer suggested, a direct analysis of ROS in mesophilic tissue will give more value to our study.

Point 2: I invite the authors to clarify a general concept; Since homeostatic modulation in defense of salt stress, at the photochemical level, is essentially manifested by an increase in NPQ, in the first 48 h while it begins to decrease at 78 h, is it possible to speak of an adaptive response only under mild stress conditions and not under severe stress? I invite the authors to establish a homeostatic parallelism of mild salt stress in photochemical processes with significant reduction of ROS and in osmoregulatory processes, where the accumulation of NaCl in vacuoles, decreases the water potential by allowing greater water recall and itself acting as an osmolyte. See Di Martino et al.2003 New Phytologist; section Osmolytes distribution in spinach leaf cells. Under severe salt stress conditions, the plant is unable to respond with either photochemical or osmolytic adaptive systems. Although the authors do not provide data on stomatal conductivity, probably under conditions of severe stress and reduced stomatal rhyme, the initiation of pseudocyclic processes with ROS formation in a photorespiratory metabolic context places the plant in a state of total physiological succumbing in response to salt stress.

Response 2: As the reviewer pointed out, an increased NPQ in the first 48 h could be regarded as an adaptive response to mild stress conditions. However, the decline in NPQ at 72 h suggests a defective thermal dissipation in the late stages of salt stress, which may be due to the lower LHCII level. Similarly, Eugenia myrtifolia plants showed an increase and a decrease in NPQ after mild and severe salt treatment, respectively (Acosta-Motos et al. 2015) [56]. In Di Martino et al. (2003) [57], spinach plants reduced photochemical capacity slightly and kept the cell osmotic balance through the production of compatible solutes under mild salt stress, whereas they were unable to respond with photochemical and osmotic adaptation under severe salt stress conditions. Our results indicate that photochemical capacity and NPQ may succumb to severe salt stress. This view has now been addressed in the Discussion section on page 14, line 413-418.

Reviewer 2 Report

The manuscript under review exemplifies high quality in both its design and execution. The researchers have meticulously crafted a study that demonstrates a thorough understanding of the subject matter. The authors present their findings with clarity and precision. The conclusions drawn from the data are well supported. This manuscript will make good contribution to the field and has good general interest.

The reviewer suggests some improvements to make this manuscript better:

11. Better clarity on plant growth configuration and sampling. Most data were obtained for “three independent experiments” but its definition is not defined in Part 2.1 (lanes 82-92). It is not clear if plant growth was conducted three times or if plants of different plots/groups were planted together at one time. It is also not clear how many plants were involved in each growth and how sampling was conducted. Was leaf sample from one single plant treated as one sample? Which leaf was sampled (the oldest, the newest?). Clarity on plant growth and sampling will be necessary to support the conclusions.

22. More comparison with previous research work: Among the comprehensive data presented, most of them are generally expected for salt response by plants. Authors should highlight the new findings over previous research work and elaborate significance. More relevant papers for rice plant salt stress and Arabidopsis should be included and compared.

33. One single cultivar was used (Oryza sativa cv. Dongjin), is this cultivar salt tolerant? What is the reason for choosing this cultivar. Ideally, two cultivars of different level of salt stress should be included to get more robust conclusion. 

Author Response

Response to Reviewer 2 Comments

Point 1: Better clarity on plant growth configuration and sampling. Most data were obtained for “three independent experiments” but its definition is not defined in Part 2.1 (lanes 82-92). It is not clear if plant growth was conducted three times or if plants of different plots/groups were planted together at one time. It is also not clear how many plants were involved in each growth and how sampling was conducted. Was leaf sample from one single plant treated as one sample? Which leaf was sampled (the oldest, the newest?). Clarity on plant growth and sampling will be necessary to support the conclusions.

Response 1: We have added an explanation for “three independent experiments” in the Materials and Methods section on page 4, line 182-183. In the section 2.12: “All data are shown as means ± standard error (SE) of nine replicates obtained from three independent experiments”. For three independent experiments, plant growth was conducted three times separately. Leaves from a few plants were collected as one sample, and the young, fully expanded leaves were harvested for sampling. In one independent experiment, each treatment of different time points (0, 24, 48, or 72 h) contains 72 plants (per each time point). This view has now been described in the Section 2.1 on page 2, line 89-93.

Point 2: More comparison with previous research work: Among the comprehensive data presented, most of them are generally expected for salt response by plants. Authors should highlight the new findings over previous research work and elaborate significance. More relevant papers for rice plant salt stress and Arabidopsis should be included and compared.

Response 2: As the reviewer pointed out, we highlighted the new findings over previous research work in the Discussion and Conclusion section. Accordingly, new references (47, 56-59, 63, and 65) for salt stress have now been added in the Discussion and Reference sections. As the reviewer commented, we made extensive revision throughout the Discussion section. We also made other minor changes accordingly throughout the text.

On page 12, line 366-367: “The enhanced production of H2O2 and MDA was used as the salt-stress indicator in various plant species [43–45]”.

On page 12, line 369-375: “NaCl stress greatly upregulated the expression of NHX1 and SOS1 (Figure 2A), key genes regulating Na+ sequestration/efflux and protecting cells from ...”

On page 14, line 413-418: “Similarly, Eugenia myrtifolia plants showed an increase and a decrease in NPQ after mild and severe salt treatment, respectively [56]. Spinach plants ...”

On page 14, line 427-429: “The levels of ALA also decreased in etiolated Brassica campestris seedlings treated with salt stress [58]. By contrast, salt stress increased ALA ...”

On page 14, line 450-page 15, line 453: “The increased level of heme in salt-stressed cucumber [59] is in support of our findings. By contrast, salt stress decreased the levels of ...”

On page 15, line 461-467. “HY1 (HO1 sub-family in Arabidopsis) is suggested to play an important role in salt acclimation signaling [65]. Expression of a B. napus HO enhances ...”

On page 15, line 480-483: “The increased activities of antioxidant enzymes, such as SOD, APX, and CAT, have also been observed with the increasing NaCl concentrations in ...”

On page 15, line 488-490: “Upon salinity stress, the Asc redox status did not significantly change in leaves of maize seedlings but decreased in root [43]. Salt stress-induced ...”

On page 16, line 511-517: “Our results demonstrate that differential regulation of chlorophyll and heme biosynthesis can contribute to protecting plants from salt-induced ...”

Point 3: One single cultivar was used (Oryza sativa cv. Dongjin), is this cultivar salt tolerant? What is the reason for choosing this cultivar. Ideally, two cultivars of different level of salt stress should be included to get more robust conclusion.

Response 3: The rice cultivar “Oryza sativa cv. Dongjin” was used for salt stress response in this study. It is not a salt tolerant or susceptible cultivar. We chose Dongjin cultivar because it is one of the cultivars widely cultivated in Korea. We agree with the reviewer’s opinion that use of two cultivars of different level of salt stress will give more robust conclusion.

Reviewer 3 Report

Please find the attachment with the detailed-corrected version of the reviewed manuscript.
In general, the manuscript "Salt Stress-Induced Modulation of Porphyrin Biosynthesis, Photoprotection, and Antioxidant Properties in Rice Plants  (Oryza sativa)" is based on results regarding physiological, biochemical and transcriptional analyses of rice seedlings subjected to salt stress and would be an interesting paper for Antioxidants readers. However, despite minor corrections in the manuscript that are labeled in the attached file, the whole discussion needs to be rewritten. The results obtained in the study have to be discussed with those obtained by other authors. Discussion must be more than just a description of the obtained results again. 

Minor editing of English language is required.

Author Response

Response to Reviewer 3 Comments 

Point 1: However, despite minor corrections in the manuscript that are labeled in the attached file, the whole discussion needs to be rewritten. The results obtained in the study have to be discussed with those obtained by other authors. Discussion must be more than just a description of the obtained results again.

Response 1: We made minor changes accordingly throughout the text as the reviewer commented. According to the reviewer’s suggestions, we made extensive revision throughout the Discussion section. Particularly, we added more comparison with previous research work in the Discussion section.

On page 12, line 366-367: “The enhanced production of H2O2 and MDA was used as the salt-stress indicator in various plant species [43–45]”.

On page 12, line 369-375: “NaCl stress greatly upregulated the expression of NHX1 and SOS1 (Figure 2A), key genes regulating Na+ sequestration/efflux and protecting cells from ...”

On page 14, line 413-418: “Similarly, Eugenia myrtifolia plants showed an increase and a decrease in NPQ after mild and severe salt treatment, respectively [56]. Spinach plants ...”

On page 14, line 427-429: “The levels of ALA also decreased in etiolated Brassica campestris seedlings treated with salt stress [58]. By contrast, salt stress increased ALA ...”

On page 14, line 450-page 15, line 453: “The increased level of heme in salt-stressed cucumber [59] is in support of our findings. By contrast, salt stress decreased the levels of ...”

On page 15, line 461-467. “HY1 (HO1 sub-family in Arabidopsis) is suggested to play an important role in salt acclimation signaling [65]. Expression of a B. napus HO enhances ...”

On page 15, line 480-483: “The increased activities of antioxidant enzymes, such as SOD, APX, and CAT, have also been observed with the increasing NaCl concentrations in ...”

On page 15, line 488-490: “Upon salinity stress, the Asc redox status did not significantly change in leaves of maize seedlings but decreased in root [43]. Salt stress-induced ...”

On page 16, line 511-517: “Our results demonstrate that differential regulation of chlorophyll and heme biosynthesis can contribute to protecting plants from salt-induced ...”

Round 2

Reviewer 3 Report

The authors corrected the manuscript according to the reviewer`s suggestions, so it is well-written and will be interesting for Antioxidants readers.

Minor editing of English language is required.